# Intracranial Hemorrhage from Dural Arteriovenous Fistulas: What Can We Find with CT Angiography?

Alberto Negro [1,*], Francesco Somma [1], Valeria Piscitelli [1], Giuseppe Maria Ernesto La Tessa [1], Carmine Sicignano [1], Fabrizio Fasano [1], Stefania Tamburrini [2], Ottavia Vargas [1], Gianvito Pace [1], Michele Iannuzzi [3], Alessandro Villa [4], Luigi Della Gatta [5], Carmela Chiaramonte [4], Ferdinando Caranci [6], Fabio Tortora [7] and Vincenzo D'Agostino [1]

[1] Department of Neuroradiology, Ospedale del Mare, Via Enrico Russo, 80147 Naples, Italy; fra1585@hotmail.it (F.S.); valeria.piscitelli@gmail.com (V.P.); glatessa@libero.it (G.M.E.L.T.); carmine.sicignano@gmail.com (C.S.); fabriziodoc@gmail.com (F.F.); ottaviavargas@gmail.com (O.V.); dr.gianvito.pace@gmail.com (G.P.); vincenzo-dagostino@libero.it (V.D.)

[2] Department of Radiology, Ospedale del Mare, Via Enrico Russo, 80147 Naples, Italy; tamburrinistefania@gmail.com

[3] Department of Anesthesia and Intensive Care, Ospedale del Mare, Via Enrico Russo, 80147 Naples, Italy; michele.ianuzzi74@gmail.com

[4] Department of Neurorsurgery, Ospedale del Mare, Via Enrico Russo, 80147 Naples, Italy; alessandrovilla83@gmail.com (A.V.); milachiaramonte87@gmail.com (C.C.)

[5] Department of Neuoradiology, AORN A.Cardarelli, Via Antonio Cardarelli, 80131 Naples, Italy; luigi.dellagatta@hotmail.it

[6] Department of Precision Medicine, University of Campania Luigi Vanvitelli, Via de Crecchio, 80138 Naples, Italy; ferdinando.caranci@unicampania.it

[7] Department of Advanced Biomedical Sciences, Federico II University Naples, 80131 Naples, Italy; fabio.tortora@unina.it

* Correspondence: Alberto.Negro@hotmail.it

**Abstract:** (1) Background: Dural arteriovenous fistulas (DAVF) represent a rare acquired intracranial vascular malformation, with a variety of clinical signs and symptoms, which make their diagnosis difficult. Intracranial hemorrhage is one of the most serious clinical manifestations. In this paper the authors' goal was to verify the accuracy and utility of contrast-enhanced brain CT angiography (CTA) for the identification and the characterization of dural arteriovenous fistulas (DAVFs) in patients who presented with brain hemorrhage compared to 3D digital subtraction angiography (3D DSA); (2) a retrospective study of 26 patients with DAVFs who presented with intracranial hemorrhage to our institution was performed. The information reviewed included clinical presentation, location and size of hemorrhage, brain CTA and 3D DSA findings; (3) results: 61% (16/26) of DAVFs were identified by CTA. The vast majority of patients were male (69%, 18/26) and the most common presenting symptom was sudden onset headache. All DAVFs had cortical venous drainage and about one-third were associated with a venous varix. The most common location was tentorial (73%, 19/26); (4) conclusions: CTA can represent a valid alternative diagnostic method to 3D DSA for the study of DAVF in the initial and preliminary diagnostic approach, especially in emergency situations. In fact, it represents a fast, inexpensive, non-invasive and above all, easily accessible and available diagnostic technique, unlike DSA or MRI, allowing to provide information necessary for the identification, classification and treatment planning of DAVFs.

**Keywords:** dural arteriovenous fistulas (DAVFs); brain CT angiography (CTA); 3D digital subtraction angiography (DSA)

## 1. Introduction

Dural arteriovenous fistulas (DAVFs) represent an infrequent acquired intracranial vascular malformation characterized by the presence of a pathological shunt located within

the dura mater of the brain [1–3] without a focal nidus, unlike many other cerebral arteriovenous malformations [1]. These vascular malformations can be identified anywhere along the dura, but are often found in the region of the transverse, sigmoid and cavernous sinuses [2,4–6]. DAVFs typically have arterial supply from the meningeal arteries, while they have venous drainage directly into the dural venous sinuses or through the cortical and meningeal veins [2,3]. Larger and more complex lesions may also have a pial arterial supply [3]. Usually, only a single lesion is found in most patients, but multiple shunts ($\leq$8%) can also be identified [7]. DAVF can present with a variety of clinical signs and symptoms in relation to the location, but especially in relation to the venous drainage pattern. The latter feature also determines the risk of serious sequelae [7–11]. Intracranial hemorrhage is one of the most serious clinical manifestations of intracranial DAVFs [6,7,12–15]. It is important in the event of this complication to quickly recognize the presence of DAVF in order to plan the best therapeutic strategy, by a surgical or endovascular approach or by a combined treatment. In this study, we verify the accuracy and the utility of contrast-enhanced brain CT angiography (CTA) for the identification and the characterization of DAVFs in patients with brain hemorrhage as first diagnostic exam after non enhanced CT exam at the emergency room of our hospital in the last five years between 2017 and 2021. We reviewed the angio CT findings comparing with those of the 3D digital subtraction angiography (3D DSA) as reference standard.

## 2. Materials and Methods

We retrospectively searched our digital archive of radiological exams for all CTA studies conducted in patients with atypical spontaneous intracranial hemorrhage in the period between January 2015 and January 2021. Only patients with both CT and CTA and subsequent 3D DSA as part of their diagnostic work-up were included in this analysis, identifying 126 patients. We focused our research on cases of atypical intracranial hemorrhage from DAVF diagnosed with 3D DSA, the gold standard diagnostic method in this kind of pathology, identifying 26 patients. Information collected included demographics (age and sex), symptoms at presentation, non-enhanced CT, CT angiography and 3D rotational angiography findings at onset, extension of hemorrhage (intraparenchymal, subdural, subarachnoid or intraventricular), location of the fistula and DAVF grade (cortical venous drainage, varix).

*CT Angiography*

CTA was performed with a Toshiba Aquilino 64-slice helical CT scanner. Volume CT was routinely performed at 130 mA and 100–120 kVp. Collimation, rotation time and pitch were optimized for the individual CT scanner according to recommendations of the manufacturer. A 90-mL dose of iodinated contrast medium (iopromide, 270 mg of iodine/mL, Visipaque 270; GE Healthcare, Cork, Ireland) was injected at a rate of 4.0 mL/s into an antecubital vein via a 20-gauge catheter, followed by 40 mL of saline solution. CT scanning was triggered by using a bolus-tracking technique, with the region of interest placed in the aortic arch. Image acquisition begins immediately after attenuation reaches the default threshold of 130–150 AU, from the aortic arch to the vertex, in order to have an opacification of the intracranial vascular circulation in the arterial phase. After 5–8 s from the first acquisition we proceeded to a second acquisition, from the vertex to the aortic arch, to have an opacification of the intracranial vascular circulation mainly in the venous phase. The images acquired in this manner allow us to evaluate both arterial and venous well-opacified vasculature and therefore allow us to evaluate both arterial and venous structures.

## 3. Dimensional Digital Subtraction Angiography (3D DSA)

Angiography was performed on a DSA operating table (Allura Xper FD20, Philips Medical, Eindoven, the Netherlands). The examination of 3D DSA was conducted in 15 non-cooperating or intubated patients (57%) under general anesthesia. Three-dimensional

rotational acquisition was conducted for both internal carotid arteries and the vertebral artery with injection of 3- to 4-mL contrast material per second. In the event that the distal contralateral vertebral artery was not visualized, an additional 2D biplane run of this vertebral artery was then performed. Angiography was followed by endovascular treatment when possible. In the other cases, a surgical therapeutic approach was planned instead.

### 3.1. Analysis of CT Angiography and of 3D Rotational Angiography

CT source images were postprocessed to create contiguous 1-mm coronal and sagittal reformations, as well as axial, coronal and sagittal MIP images with both 3- and 5-mm section thickness. Volume-rendered 3D images and curved planar reformatted images of the bilateral common, internal and external carotid arteries and vertebral arteries were also created, as well as 3D images of the vessels of the circle of Willis. The following angio CT findings were evaluated and assigned to the patients examined: either asymmetric, dilated feeding arteries, or both; numerous, asymmetric, dilated and engorged cortical veins; "sinus findings" ("shaggy" appearance of a dural venous sinus or the tentorium cerebella, asymmetric attenuation of venous sinus and dural sinus thrombosis); and transcalvarial channels [16,17]. Source images, post-processed images and 3D reconstructions with bone removal were transferred to a picture archiving and communication system (PACS). Native CT and CTA data were collected in a dedicated work list in PACS. Neuroangiographic datasets were evaluated on the same PACS workstation used for the prior analysis. The presence of abnormal connections between arterial feeders and a dural venous sinus or leptomeningeal vein constituted diagnostic criteria for DAVF. Fistulas were categorized on the basis of their location and the presence of venous cortical drainage. If the vein draining the intradural nidus emerged from the upper or lower tentorial surface and also from the area of the tentorial incisura, including the galenic system and the tentorial attachment, then the fistula was localized to the tentorium. For this classification criterion we have included fistulas that drain through the petrous complex in "tentorial" DAVFs, as lateral tentorial fistulas, rather than including them in superior petrosal sinus fistulas according to the classification system of Lawton et al. [6]. All DAVFs were classified according to the Cognard stage on the basis of conventional angiography [8].

### 3.2. Statistical Analysis

We evaluated the diagnostic value of the CTA in the selected study population sample using a contingency table and the specific functions of the data analysis tool provided by a commonly used software. We conducted a post hoc sample size calculation, with a binomial dichotomous primary endpoint, conducted with a common statistical software available online, dividing the population of 26 subjects affected by DAVF into the groups of those with CTA positive findings and those without CTA positive findings to better establish the predictive value of the CTA results identified in our study.

## 4. Results

The most frequent onset clinical symptomatology was represented by headache, present in 73% of cases and most patients were male (76%) with a mean age of 55 years. Other patients showed altered metal status (57%) and seizures (3%). The most frequent localization of DAVF was tentorial (57%). All fistulas showed cortical venous drainage (Cognard IIb or greater) with a venous varix (Cognard grade IV) in 38% of subjects. Intraparenchymal hemorrhage (76%) was the most frequently identified form of bleeding in our series, often located in a position of contiguity to the fistula rather than to the actual and current point of fistulization. Other types of bleeding included intraventricular hemorrhage (46%) and subdural hemorrhage (19%). All these results regarding the demographic data, the clinical symptoms of onset, the localization of the fistula and the type and extent of intracranial hemorrhage are shown in Table 1.

**Table 1.** Demographics, location of fistula, and extent of presenting hemorrhage.

| Parameter | Value |
|---|---|
| No. of patients | 26 |
| Mean age in years | 55 |
| Male | 20 |
| Presenting symptom/sign: | |
| ● Headache | 19 |
| ● altered mental status | 15 |
| ● seizures | 5 |
| DAVF grade | |
| ● cortical venous drainage | 26 |
| ● varix | 10 |
| Location of fistula: | |
| ● tentorial | 15 |
| ● transverse sinus | 5 |
| ● other | 6 |
| Hemorrhage: | |
| ● intraparenchymal | 20 |
| ● intraventricular | 12 |
| ● subdural | 5 |

Based on the results of 3D DSA, in relation to the Cognard classification [8], 5 patients showed a fistula with retrograde flow within the sinus or a fistula with retrograde reflux into cortical veins (type IIa, type IIb), 12 patients a fistula with direct drainage into cortical veins (type III) and 10 patients a fistula with reflux into ectasic cortical veins (type IV), Table 2.

**Table 2.** Subdivision of the patients in relation to the Cognard classification *.

| Cognard Type | No. of Patients |
|---|---|
| I | 0 |
| IIa | 0 |
| IIa + b | 5 |
| III | 11 |
| IV | 10 |
| V | 0 |

* Cognard classification [8].

Brain CT angio studies allowed the identification of the fistula in 16/26 (61%) patients, with a diagnostic sensibility of 61%, a diagnostic specificity of 100% and a diagnostic accuracy of 92 % (Table 3). The post hoc sample size's statistical power was 20%.

**Table 3.** Contingency table of CTA diagnostic value.

| | DAVF+ | DAVF− |
|---|---|---|
| CTA+ | 16 | 0 |
| CTA− | 10 | 100 |

CTA+: patients with positive CTA findings of DAVF presence; CTA−: patients without CTA positive findings of DAVF presence; DAVF+: patients with DAVF (with a diagnosis confirmed by 3D DSA); dAVF−: patients without DAVF (without a diagnosis confirmed by 3D DSA).

Most of the patients with CTA positive findings of DAVF presence, confirmed by the subsequent 3D DSA study, showed numerous and engorged cortical veins (Figures 1 and 2) and asymmetric attenuation of venous sinus (Figure 2). The CTA findings in the group of patients with DAVF are summarized in Table 4.

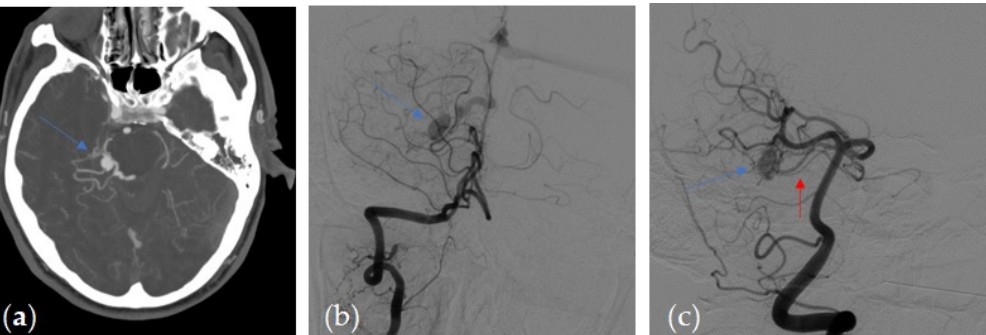

**Figure 1.** (**a**) Axial 3-mm-slab MIP from CTA. Dilated engorged cortical veins with a prominent dilated vein (blue arrow) in a patient with a right tentorial DAVF (**b**,**c**) DSA, arterial phase, right vertebral artery injection in antero-posterior (**b**) and in oblique projection (**c**) in the same patient 1 shows dilated arterial feeders from right superior cerebellar artery (red arrow) and engorged cortical veins, that correlate with the angio CT findings.

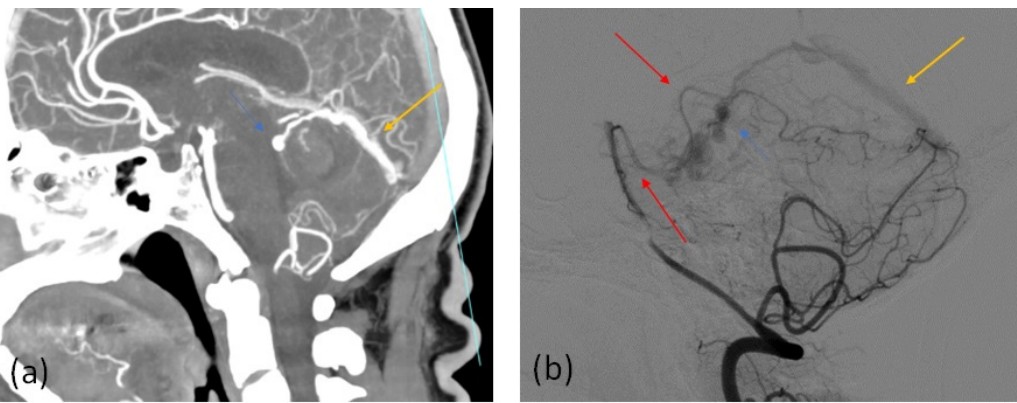

**Figure 2.** (**a**) Sagittal 3-mm-slab MIP from CTA. Dilated engorged cortical veins with a prominent dilated vein (blue arrow) in the patient 1 with a right tentorial DAVF with abnormal early opacification of the straight sinus (orange arrow), showing arterial contrast intensity, differently from the other intracranial venous sinus (**b**) DSA, arterial phase, right vertebral artery injection in the sagittal projection in the same patient 1 shows arterial feeders from the right superior cerebellar artery, enlarged cortical veins with a prominent vein (blue arrow) and reflux in the straight sinus (orange arrow), the correlate of the arterial contrast intensity shown on CTA in (**a**), proving the arterialization of venous structures. The other venous sinus did not show the same attenuation.

**Table 4.** CTA findings in the group of patients with DAVF.

| Brain Angio CT Positive Findings | No. of Patients |
|---|:---:|
| Asymmetric and/or dilated feeding arteries | 7 |
| Numerous and engorged cortical veins | 15 |
| Sinus findings: | |
| • "Shaggy" appearance of a dural venous sinus or the tentorium cerebella | 8 |
| • Asymmetric attenuation of venous sinus | 10 |
| • Dural sinus thrombosis | 4 |
| Transcalvarial channels | 4 |

## 5. Discussion

In our series of patients with DAVF intracranial hemorrhage, the most frequent form of bleeding was parenchymal hemorrhage, the epicenter of which was located closer to the fistula rather than further away from the actual point of fistulization. The position, the arterial supply, the degree of arteriovenous shunt, and above all, the type of venous drainage, determine the clinical manifestations of DAVF in a wide and varied spectrum [1,3,7]. DAVF without cortical venous drainage (CVD) may be asymptomatic or may present with symptoms of increased blood flow in the dural sinus, such as pulsatile tinnitus, particularly common in the case of lesions involving the transverse and sigmoid sinuses [1,2,8,18–20]. Generalized neurological symptoms may be due to venous hypertension or cerebrospinal fluid malabsorption, while other symptoms, such as cranial nerve palsy, often refer to arterial steal or mass effect events caused by an enlarged arterial supply [3,7,21]. Furthermore, if DAVF involves the cavernous sinus, the patient may present severe orbital symptoms, such as chemosis, proptosis and ophthalmoplegia, up to a drastic reduction in visual acuity [1–3,19,22]. In the case of DAVF with CVD, there may be much more serious clinical presentations, such as various types of intracranial bleeding that are accompanied by severe clinical symptoms, characterized, for example, by severe and sudden headache, convulsions and loss of consciousness [3,5,18,23]. The presence of DAVF complicated by retrograde leptomeningeal drainage in either the cerebral, cerebellar cortical veins, or both, is a dangerous condition because it may cause intracranial hypertension and increase the risk of intracranial hemorrhage (ICH). Since these veins pass through various compartments, various types of bleeding can occur in the event of their rupture, such as subdural, subarchnoid and cerebral hemorrhage. At the rupture site we can often identify a focal ectasia of abnormal draining veins. Retrograde drainage from a shunt directly into the leptomeningeal vein is associated with a higher risk of ICH than retrograde drainage directly into the dural sinus. Hemorrhage is usually intraparenchymal, less frequently subdural and subarchnoid. Since all intracranial DAVFs can be associated with leptomeningeal venous reflux, they can then present with ICH. However, some DAVFs with tentorial or anterior cranial fossa locations are associated more or necessarily with retrograde leptomeningeal venous reflux due to local venous anatomy and therefore may occur more frequently with ICH (in 75–95% of cases) [3]. According to Lasjaunias et al., there is an important correlation between focal neurological deficit and CVD or venous congestion in the affected vascular territory [7]. However, there may be less frequent severe clinical presentations, such as brain stem or cerebellar dysfunction secondary to venous congestion, Parkinsonism-like symptoms, extra-axial hemorrhage of the cervical spine, as well as cervical and upper thoracic myelopathy [3,7,21,24]. DAVF, especially if accompanied by a very extensive arteriovenous shunt and dural sinus thrombosis, can induce impaired venous drainage from the brain with consensual global venous hypertension [3,25]. This condition can lead to cerebral edema, encephalopathy and cognitive decline [3,25–28]. This varied possibility of clinical presentations can make the diagnosis of DAVF very difficult. In patients with unexplained intracranial hemorrhage or neurological deficit, the clinician should suspect the presence of DAVF with a high probability. The initial radiological evaluation is essentially based on cross-sectional non-contrast computed tomography (CT) and magnetic resonance imaging (MRI) studies. However, non-contrast computed tomography (CT) and conventional magnetic resonance (MR) show often irrelevant diagnostic accuracy with benign DAVF [29]. However, the sequelae of aggressive lesions with CVD can be evaluated in these studies. In these cases, not only hemorrhage but also venous congestion with edema and venous aneurysms or with tortuous cortical veins can be identified up to the formation of a pseudophlebitic pattern with parenchymal or leptomeningeal enhancement [29–31]. Brain CT alone does not allow the diagnosis of DAVF, but it allows detection of vasogenic edema due to venous congestion and allows identifying of intracranial hemorrhage, which is usually far from the localization of the arterial venous shunt since the varices venous veins often represent the source of the bleeding itself. [32]. CT or MRI may show prominent or hollow flow vessels associated with a dural sinus in the case

of larger and more extensive fistulas [1,29–31]. CT or MRI angiography (CTA and MRA, respectively) can be used to identify the presence of DAVF, evaluate their location and indicate a certain degree of classification, offering useful information to understand the risk of bleeding and also to evaluate the response to treatment [17,29,30,33]. Both CTA and MRA may visualize the fistula itself as prominent vessels associated with the meninges or dural sinus wall, as well as detect enlarged feeding arteries, early dural sinus opacification and prominent draining veins [17,29,30,33]. In our study, 7 patients showed either asymmetric, dilated feeding arteries, or both, with CTA with a large overlap on the arterial supply pattern found on 3D DSA. Fifteen patients showed numerous, asymmetric dilated and engorged cortical veins (Figure 1). Our results confirm that most DAVFs associated with CVD demonstrate either dilated leptomeningeal, medullary vessels, or both, without an identifiable nidus [26,30,34,35]. These generally corresponded to the anatomical location of the CVD on DSA. These dilated and tortuous vessels perhaps constitute a network of CVD-involved veins and a collateral venous circuit, through which the deoxygenated blood is redirected away from the normal venous drainage pathways, which are instead functionally obstructed, as shown by the pseudophlebitic pattern on DSA [31] and further supported by a SWI series [36]. The presence of CVD is a negative prognostic angiographic finding. It is important to try to establish the type of venous drainage. If the direction of drainage is antegrade, then the clinical course is more favorable. On the other hand, retrograde reflux predisposes to an increase in their regional intracranial pressure that can increase the hemorrhagic risk. It was not possible to identify the correct type of venous drainage, the precise position of the fistula points and the direction of flow within the structures of the venous sinus, using the CTA, unlike the 3D DSA, which is the gold standard diagnostic method for correctly identifying and classifying DAVFs. The failure to obtain time-resolved projection images in the CTA makes it difficult or impossible to correctly represent the exact anatomy of a DAVF and its type of venous drainage. The CTA does not allow for analyzing the directionality of the blood flow, but we believe that a cortical venous dilatation identifiable at the CTA can be considered a valid correlation, especially if it is associated with the radiological finding of arterial enhancement of a cortical venous structure during the arterial phase of image acquisition with CTA, which essentially represents the arterialization of a cerebral vein. A portion of patients with DAVFs demonstrated "sinus findings". Eight subjects showed a "shaggy" appearance of the sinus or tentorium. This "shaggy" appearance constitutes a set of pathological sequelae involved in the angiogenic process, where various factors (represented for example by the opening of the dural vascular channels, thickening and intimal stenosis, venous hypertension and dilatation and reflux in the cortical veins) contribute to this aspect. Ten patients showed local or regional differences in contrast enhancement of the dural sinus (mostly transverse and sigmoid sinus) with higher contrast intensity compared with that of the contralateral corresponding venous structures. Therefore, the venous sinus with greater intensity showed an improvement in contrast comparable to that of the cerebral arteries depicted, demonstrating the arterialization of the venous structures (Figure 2). The modern multidetector CT (MDCT) allows obtaining more sections during a sufficiently short time interval to allow for detection of even very small variations in the intensity of the contrast. Using late arterial phase MDCT angiography scan with low intensity of venous structures, arterial venous blood can be detected within the cerebral venous vessels if there is a draining DAVF in a dural sinus. For this reason, it is very important to conduct the inspection of the source images according to exact and narrow window settings, so as not to lose even small differences in the intensity of the contrast between closely related vascular structures within a single layer. Four patients had dural sinus thrombosis. Presence of veno-occlusive disease of the major dural venous sinuses was encountered in 15% of the patients, differently from prior reports with reported incidences ranging from 39 to 68% [17,37,38], probably due to the low sample size. The venous anomalies found were localized in proximity to DAVFs or in their downstream pathways as in a previous study [38]. There have been concomitant episodes of acute vein thrombosis at the time of diagnosis of DAVF. Thrombosis of a dural venous

sinus represents one of the main factors contributing to the development of a DAVF [38], while, on the contrary, thrombosis of the DAVF itself or one of its outlets, can contribute to the spontaneous conversion to aggressive fistula or inversion to benign type. In the case of a venous thrombosis, especially if it obstructs the venous outflow, phenomena of venous stasis or bidirectional venous flow can occur. They contribute to the precipitation of other venous thrombotic events by inducing significant impairment of the global local venous drainage, thus causing venous infarcts or hematomas. Similarly, the presence of spontaneous DAVF thrombosis could also explain the bleeding lesions in patients without CVD observed in our series [7,26,38]. Four subjects had transcalvarial channels, which represent the path of the relative transosseous arterial feeders. They originated from the meningeal branches of the occipital artery and were therefore located in the occipital bone around the occipito-mastoid suture [39]. A study did not find high sensitivity for this sign [17]. The alteration of the blood flow dynamics, in many cases, represents the main anomalous finding of DAVF. Imaging techniques with non-invasive static methods, such as conventional CT scans with contrast medium, conventional MRI images or even CT angiography and time-of-flight MRI angiograms, have limited diagnostic value and in many identifications of DAVFs can be very difficult and represent a diagnostic challenge in many cases. Narvid et al. [17] were able to identify the presence of arterial feeders associated with 86% sensitivity and 100% specificity in a group of patients with pulsatile tinnitus using the CTA technique. Similarly, Kwon et al. [30] identified an abnormal increase related to venous flow using time-of-flight magnetic resonance angiography (MRA) with a sensitivity of 91% (*n* = 11). In our study, we were able to identify the presence of DAVF in 76% of patients with intracranial hemorrhage with CTA. The technological progress of recent years has allowed time-resolved CTAs and MRAs, which are able to trace the passage of a contrast bolus through cerebral vessels [33,39–41]. Technological progress in recent years has allowed time-resolved CTAs and MRAs, capable of tracing the passage of a contrast bolus through cerebral vessels [33,39–41]. Willem et al. [39] successfully detected and classified 90% of a small group of DAVFs (*n* = 11) with time-resolved CT angiography. Similarly, Meckel et al. [40] demonstrated that time-resolved dynamic MRA allowed detection of the presence and lateralization of all DAVFs without false positives. Pekkola et al. [33] found a higher sensitivity (94.4%) and specificity (83.3%) than the time-resolved MRA technique (sensitivity 64.7% and specificity 80%) for the detection of arteriovenous shunting in a small group of posterior fossa DAVF patients (*n* = 19). Despite these advances in MR and CT imaging, catheter angiography remains the definitive imaging study for evaluation of DAVF because of its superior spatial and temporal resolution [1,40]. Angiography with a catheter allows identifying a series of very important characteristics to establish the risk of bleeding, by delineating both the arterial supply and the venous drainage of the fistula and to identify CVD, with possible obstruction of the venous outflow, the pedicle arterial and possible concomitant venous aneurysms. Catheter angiography is also an excellent diagnostic technique for evaluating any associated dural venous sinus thrombosis or occlusion. Finally, all the information that catheter angiography can provide is essential for endovascular or surgical treatment planning [40]. However, since DAVFs present with very variable symptoms and since angiography is an invasive and expensive diagnostic technique, non-invasive diagnostic methods are interesting. The CTA shows a number of direct and indirect signs that suggest intracranial DAVF. These signs can be used to screen for DAVF in ICH patients. In particular, such as in our series, when we find the presence of numerous and engorged cortical veins and asymmetric attenuation of venous sinus with a brain CT angio exam, we can suspect the presence of DAVF with a high probability, mostly in a patient with an atypical ICH.

*Limitations of the Study*

The major limitations of our study are represented by the low sample size with a low post hoc sample size statistical power (20%) and the retrospective design. In fact, these limits do not allow us to understand the predictive value of the CTA findings that

we identified in our study regarding the presence of a DAVF. However, the evaluation of the sensitivity, specificity and diagnostic accuracy of the CTA in our study were carefully analyzed, confirming the high diagnostic specificity and accuracy of CTA in DAVF identification. Future studies with a higher sample size are needed to better evaluate the sensitivity, specificity and diagnostic accuracy of the brain CTA in the identification of DAVF and to validate the predictive value of the radiological findings at a CTA exam in this type of vascular malformation.

## 6. Conclusions

CTA represents a diagnostic technique readily available and accessible 24 h a day, compared to DSA and magnetic resonance imaging (MRI), allowing, in many cases, to offer useful information to identify and classify a DAVF. However, its sensitivity is reduced compared to 3D DSA, especially in identifying small or low-flow fistulas. Larger studies are needed to evaluate the sensitivity and specificity of this diagnostic method for the study of DAVFs.

**Author Contributions:** Conceptualization, A.N., F.S., A.V., F.T., F.C. and V.D.; methodology, A.N, F.S., V.D., V.P., F.F., G.M.E.L.T., O.V., C.C., M.I. and G.P.; software, AN., F.S., V.D., V.P., C.S., O.V., F.S., S.T. and L.D.G.; validation, A.N., F.T. and F.C.; formal analysis, A.N., A.V., F.S., F.T., F.C, S.T., M.I. and G.P.; investigation, A.N., F.S., V.D., V.P., C.S., O.V., F.S., S.T. and L.D.G.; data curation, S.T., A.N. and F.S.; writing—original draft preparation, A.N., F.S., S.T. and F.C.; writing—review and editing, A.N., F.S., A.V., C.C. and S.T.; supervision, F.T. and F.C. All authors have read and agreed to the published version of the manuscript.

**Funding:** This research received no external funding.

**Institutional Review Board Statement:** No need to request ethical review and approval for this study because it is based on the retrospective analysis of results from clinical radiological care activities.

**Informed Consent Statement:** Informed consent was obtained from all subjects involved in the study.

**Data Availability Statement:** The data presented in this study are available within the presented and described article. Further data can be requested from the corresponding author.

**Conflicts of Interest:** The authors declare no conflict of interest.

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
