# Peer review of "Intracranial Hemorrhage from Dural Arteriovenous Fistulas: What Can We Find with CT Angiography?"

_tomography, doi:10.3390/tomography7040068_

Round 1

Reviewer 1 Report

Simple case serie reporting the diagnostic accuracy of CTA and DSA in the  diagnosis of dAVF. 

The superior role of DSA in this disease is well known, the current study does not add any relevant information on this aspect. 

The design of the study is simple. 

Introduction: nothing to concern. 

M&M:  CT angiography section should be better explained. Single phase, multiphase, timing?

I suggest to add a statistical study in order to strength the results of the study.

Results: too simple description. Quality of the provided images is good.

Discussion: too long, even if critical in many aspects. I would add some lines regarding the relevance of bleeding pattern on the detection of dAVF with CTA.

References: ok

Author Response

-Simple case serie reporting the diagnostic accuracy of CTA and DSA in the diagnosis of dAVF. The superior role of DSA in this disease is well known, the current study does not add any relevant information on this aspect. The design of the study is simple. I suggest to add a statistical study in order to strength the results of the study

In the submitted paper we describe an observational case series retrospective study where we analyzed the images of brain CTA exams in patients with spontaneous intracranial hemorrhage from dAVF in order to evaluate this diagnostic technique in this pathological condition, having as a reference the 3D DSA, the gold standard diagnostic method for the study of dAVF. We searched our digital archive of radiological exams for all CTA studies conducted in patients with atypical spontaneous intracranial hemorrhage in the period between January 2015 and January 2021. We identified 126 patients. All patients underwent a 3D DSA study, the gold standard reference method in this kind of pathological condition, both in the cases of positive and negative results for cerebral vascular malformations in the previous CTA study, as per our clinical neuroradiological assistance and therapeutic protocol in case of spontaneous intracranial hemorrhage. We focused our research on cases of atypical intracranial hemorrhage from DAVF diagnosed with 3D DSA and we found 26 patients. CTA allowed the identification of the fistula in 16/26 (61%) patients, with 61% of diagnostic sensibility, 100% of specificity and 92 % of accuracy. I report below the contingency table that was used to determine the diagnostic parameters described above, calculated with specific statistical formulas conducted with common data analysis software.

dAVF  +

dAVF -

CTA +

16

0

CTA -

10

100

We reviewed positive CTA studies for the presence of dAVF and evaluated the following radiological findings, asymmetric and/or dilated feeding arteries; numerous, asymmetric, dilated and engorged cortical veins; “sinus findings” («shaggy» appearence of a dural venous sinus or the tentorium cerebella, asymmetric attenuation of venous sinus and dural sinus thrombosis); transcalvarial channels, in order to evaluate those that occurred in patients with greater frequency and therefore could be considered as the most suggestive and predictive radiological findings of dAVF.

-Introduction: nothing to concern. 

-M&M:  CT angiography section should be better explained. Single phase, multiphase, timing?

CTA was performed with a Toshiba Aquilino 64-slice helical CT scanner. Volume CT was routinely performed at 130 mA and 100–120 kVp. Collimation, rotation time and pitch were optimized for the individual CT scanner according to recommendations of the manufacturer. A 90-mL dose of iodinated contrast medium (iopromide, 270 mg of iodine/mL, Visipaque 270; GE Healthcare, Cork, Ireland) was injected at a rate of 4.0 mL/s into an antecubital vein via a 20-gauge catheter, followed by 40 mL of saline solution. CT scanning was triggered by using a bolus-tracking technique, with the region of interest placed in the aortic arch. Image acquisition begins immediately after attenuation reaches the default threshold of 130– 150 AU, from the aortic arch to the vertex, in order to have an opacification of the intracranial vascular circulation in the arterial phase. After 5-8 seconds from the first acquisition we proceed to a second acquisition, from the vertex to the aortic arch, to have an opacification of the intracranial vascular circulation mainly in the venous phase.

The images acquired in this manner allow us to evaluate both arterial and venous vasculature well-opacified and therefore allow us to evaluate both arterial and venous structures. CT source images were postprocessed to create contiguous 1-mm coronal and sagittal reformations, as well as axial, coronal, and sagittal MIP images with both 3- and 5-mm section thickness. Volume-rendered 3D images and curved planar reformatted images of the bilateral common, internal and external carotid arteries and vertebral arteries were also created, as well as 3D images of the vessels of the circle of Willis.

-Results: too simple description. Quality of the provided images is good.

The results have been described in a more exhaustive and complete way, adding the paragraph regarding the diagnostic value of the CTA and inserting in this regard the specific contingency table that we used to determine the diagnostic sensitivity, specificity and accuracy of this radiological method.

-Discussion: too long, even if critical in many aspects. I would add some lines regarding the relevance of bleeding pattern on the detection of dAVF with CTA.

I include the required lines in the discussion. I report them below. I made the discussion more fluid.

The presence of dAVF complicated by retrograde leptomeningeal drainage in the cerebral and / or cerebellar cortical veins is a dangerous condition because it may cause intracranial hypertension and increase the risk of intracranial hemorrhage (ICH). Since these veins pass through various compartments, various types of bleeding can occur in the event of their rupture, such as subdural, subarchnoid and cerebral hemorrhage. At the rupture site we can often identify a focal ectasia of abnormal draining veins. Retrograde drainage from shunt directly into the leptomeningeal vein is associated with a higher risk of ICH than retrograde drainage directly into the dural sinus. Hemorrhage is usually intraparenchymal, less frequently subdural and subrachnoid. Since all intracranial dAVFs can be associated with leptomeningeal venous reflux then they can present with ICH. However, some dAVFs with tentorial or anterior cranial fossa locations are associated more or necessarily with retrograde leptomeningeal venous reflux due to local venous venous anatomy and therefore may occur more frequently with ICH (in 75-95% of cases) [Sarma D, ter Brugge K. Management of intracranial dural arteriovenous shunts in adults. Eur J Radiol. 2003;46:206–220].  It is therefore important to try to establish the type of venous drainage. If the direction of drainage is antegrade then the clinical course is more favorable. On the other hand, retrograde reflux predisposes to an increase in their regional intracranial pressure that can increase the hemorrhagic risk. As described in our study, it was not possible to identify the correct type of venous drainage, the precise position of the fistula points and the direction of flow within the structures of the venous sinus, using the CTA, unlike the 3D DSA which is  the gold standard diagnostic method for correctly identifying and classifying DAVFs. The failure to obtain time-resolved projection images in the CTA makes difficult or impossible to represent correctly the exact anatomy of a DAVF and the type of its venous drainage. Even if the CTA does not allow to analyze the directionality of the blood flow, we believe that a cortical venous dilatation identifiable at the CTA can be considered a valid correlation, especially if it is associated with the radiological finding of enhancement of a cortical venous structure during the arterial phase of image acquisition with CTA, which essentially represents the arterialization of a cerebral vein.

-References: ok

Reviewer 2 Report

  • the small sample size is a limitation of the study. This should be discussed in a dedicated limitation section.
  • a post-hoc sample size calculation should be provided.
  • the retrospective nature of this study is a limitation. This should be discussed in a dedicated limitation section

Author Response

-the small sample size is a limitation of the study. This should be discussed in a dedicated limitation section; the retrospective nature of this study is a limitation. This should be discussed in a dedicated limitation section

I have described the limitations of our study in a separate paragraph. I report it below.

The major limitations of our study are represented by the low sample size with a low post hoc sample size statistical power and the retrospective design. In fact, these limits do not allow us to understand the predictive value of the CTA findings of the that we identified in our study regarding the presence of a DAVF. However, the evaluation of the sensitivity, specificity and diagnostic accuracy of the CTA in our study were carefully analyzed and the post-hoc sample size calculation conducted with a common statistical software available online showed a post-hoc power of 20%.

Future studies with a higher sample size are needed to better evaluate the sensitivity, specificity and diagnostic accuracy of the brain CTA in the identification of dAVF and to validate the predictive value of the radiological findings at a CTA exam in this type of vascular malformation.

-a post-hoc sample size calculation should be provided.

I report below the results of the suggested post-hoc sample size calculation conducted with a common statistical software available online. We conducted a post-hoc sample size calculation, with a binomial dichotomous primary endpoint, conducted with a common statistical software available online, dividing the population of 26 subjects affected by DAVF into the group of those with CTA postive findings and those without CTA positive findings to better establish the predictive value of the CTA results identified in our study.

Round 2

Reviewer 1 Report

The manuscript has been significantly improved

Reviewer 2 Report

the authors well addressed my previous comments. The paper improved very much